# Leveraging Functional Genomics for Understanding Beef Quality Complexities and Breeding Beef Cattle for Improved Meat Quality

**DOI:** 10.3390/genes15081104

**Published:** 2024-08-22

**Authors:** Rugang Tian, Maryam Mahmoodi, Jing Tian, Sina Esmailizadeh Koshkoiyeh, Meng Zhao, Mahla Saminzadeh, Hui Li, Xiao Wang, Yuan Li, Ali Esmailizadeh

**Affiliations:** 1Inner Mongolia Academy of Agricultural & Animal Husbandry Sciences, Hohhot 010031, China; tianj729@163.com (J.T.); 13904711259@163.com (M.Z.); lihuizh@126.com (H.L.); xiao19870629@126.com (X.W.); liyuannmg2018@163.com (Y.L.); 2Department of Animal Science, Faculty of Agriculture, Shahid Bahonar University of Kerman, Kerman P.O. Box 76169-133, Iran; maryam.mahmoodi@agr.uk.ac.ir (M.M.); sinaesmaili2@gmail.com (S.E.K.); saminmahla@gmail.com (M.S.); aliesmaili@uk.ac.ir (A.E.)

**Keywords:** beef cattle, functional genomics, genomic selection, GWAS, meat quality, molecular breeding, omics technologies

## Abstract

Consumer perception of beef is heavily influenced by overall meat quality, a critical factor in the cattle industry. Genomics has the potential to improve important beef quality traits and identify genetic markers and causal variants associated with these traits through genomic selection (GS) and genome-wide association studies (GWAS) approaches. Transcriptomics, proteomics, and metabolomics provide insights into underlying genetic mechanisms by identifying differentially expressed genes, proteins, and metabolic pathways linked to quality traits, complementing GWAS data. Leveraging these functional genomics techniques can optimize beef cattle breeding for enhanced quality traits to meet high-quality beef demand. This paper provides a comprehensive overview of the current state of applications of omics technologies in uncovering functional variants underlying beef quality complexities. By highlighting the latest findings from GWAS, GS, transcriptomics, proteomics, and metabolomics studies, this work seeks to serve as a valuable resource for fostering a deeper understanding of the complex relationships between genetics, gene expression, protein dynamics, and metabolic pathways in shaping beef quality.

## 1. Introduction

Beef quality is defined by several traits that influence the eating experience and desirability of the meat. Key traits include palatability factors such as tenderness, juiciness, and flavor, which directly impact consumer satisfaction [1]. Tenderness refers to the ease of chewing and breaking down the meat, while juiciness is the moisture released during mastication [2]. Flavor encompasses the combined sensations of taste and aroma that make the meat appealing [3]. Visual characteristics like meat color, fat color, and marbling also play a crucial role in perceived quality [4]. The bright, desirable lean color, white fat color, and intramuscular fat distribution (marbling) enhance the appearance and contribute to flavor and juiciness [5]. Other traits like water-holding capacity, pH, and intramuscular fat content further influence overall quality, shelf life, and sensory properties [6,7]. This complex array of beef quality traits is shaped by the relationships between genetic factors, breed influences, nutrition, management practices, and post-harvest handling procedures [8]. These factors make beef quality a challenging target for traditional breeding strategies. However, the availability of high-quality bovine genome assemblies coupled with the advent of high-throughput sequencing technologies has paved the way for the integration of omics technologies, encompassing genomics, transcriptomics, proteomics, and metabolomics, to unravel the complex mechanisms underlying beef quality.

The multi-omics approach has the power to resolve the meat quality research into an image of what is being expressed, translated, and produced, which incorporates technologies characterizing various biological products, including DNA (genomics), RNA (transcriptomics), protein (proteomics), and metabolites (metabolomics) in biological samples. Genomic approaches, such as genome-wide association studies (GWAS) and genomic selection (GS), have provided valuable insights into the genetic architecture of beef quality traits. While these methods have identified several genetic markers and regions associated with these traits, their practical application in enhancing beef quality through selective breeding remains limited and requires further research and validation. These techniques enable the identification and utilization of functional variants associated with desirable phenotypes, thereby accelerating genetic improvement and enhancing the efficiency of breeding programs. Transcriptomics, which examines the expression patterns of genes, provides insights into the molecular pathways and regulatory networks governing muscle development, metabolism, and meat quality attributes. Proteomics, on the other hand, offers a comprehensive view of the functional proteins involved in these processes, elucidating their roles and interactions. Complementing these approaches, metabolomics unveils the complex metabolic landscapes that shape the biochemical composition and sensory properties of beef. This article aims to summarize the latest findings from these advanced scientific approaches in beef quality genetics. By exploring research from GWAS, genomic selection, transcriptomics, proteomics, and metabolomics, we seek to provide a comprehensive understanding of how genetic factors, gene expression, proteins, and metabolic processes influence beef quality. The goal is to offer valuable insights for researchers and industry professionals, potentially improving breeding strategies and production methods to enhance beef quality.

## 2. Functional Mutations and Commercialized DNA Tests for Beef Quality

At the beginning of the genomics era during the 1980s, the primary application of this technology in livestock breeding revolved around developing standalone genome marker tests, particularly for identifying inherited diseases and parentage testing. However, as the field progressed, the focus shifted towards integrating quantitative and genomic approaches to identify genomic variants with substantial effects on desirable traits of interest. These DNA tests were then leveraged in breeding programs, enabling breeders to make more informed decisions by selecting animals with favorable genetic profiles for specific traits, thereby accelerating genetic improvement in livestock populations.

Kostusiak et al. [9] provided a comprehensive review of the effects of single nucleotide polymorphisms (SNPs) in four key genes—myostatin (*MSTN*), thyroglobulin 5 (*TG5*), μ-calpain (*CAPN1*), and calpastatin (*CAST*)—on beef cattle productivity and meat quality traits. MSTN is a negative regulator of muscle growth. Inactivating mutations or suppression of the *MSTN* gene leads to a “double-muscled” phenotype with increased muscle mass and reduced fat deposition in cattle breeds like Belgian Blue and Piedmontese [10]. Meat from MSTN-null cattle exhibits improved tenderness across all cuts, including typically tougher cuts like chuck and round. This is likely due to increased muscle fiber hyperplasia rather than just hypertrophy [11]. While inactivating the *MSTN* gene can dramatically increase muscle yields and tenderness in beef cattle, it comes at the cost of reduced marbling and juiciness. An optimal approach leverages *MSTN* alongside other genes to strike a balance between production efficiency, leanness, and eating quality traits like tenderness and flavor. Esmailizadeh et al. [12] investigated the effects of a specific single nucleotide polymorphism (SNP) in the myostatin (*MSTN*) gene, resulting in a phenylalanine to leucine substitution at position 94 (F94L), on various beef production and quality traits. The F94L variant of *MSTN* was found to provide a more desirable intermediate phenotype than the severe double-muscling caused by complete *MSTN* inactivation, offering improved meat yield while maintaining acceptable meat quality traits like tenderness [12].

Polymorphisms in the thyroglobulin (*TG5*) gene can significantly impact beef quality, particularly in terms of intramuscular fat (IMF) content and marbling. The *TG5* gene is located on bovine chromosome 14 and encodes the thyroglobulin protein, which plays a role in fat metabolism. Wood et al. [13], in their meta-analysis, found that there was a positive association between the polymorphic forms of *TG5* and the degree of meat marbling. A specific single nucleotide polymorphism (SNP) in the 5′ untranslated region of *TG5*, characterized by a C > T transition at position −422 (X05380.1:g.−422 C > T), has been widely studied [9]. The *TG5* C allele has been associated with higher levels of IMF and increased marbling scores in beef cattle across multiple breeds [13]. Highly marbled beef, particularly from breeds such as Hanwoo and Wagyu, is valued for its tenderness, flavor, and overall palatability, which are significantly influenced by intramuscular fat (IMF) and marbling characteristics [14,15]. However, some consumers, especially those in developed countries, prefer leaner beef with a lower fat content for health reasons, creating a conflict with the preference for marbled, flavorful meat in blind taste tests. Research indicates that higher levels of IMF, especially those enriched with monounsaturated fatty acids like oleic acid, enhance sensory attributes such as juiciness and flavor, contributing to consumer preference in markets like the U.S., Japan, and Korea [14]. Studies show that marbling fleck characteristics, including their size and distribution, impact the sensory quality of beef, with finer marbling generally preferred over coarser types [16]. Additionally, the genetic predisposition of certain breeds, such as the Japanese Black, to produce high marbling levels positions them as premium products, aligning with findings that suggest consuming such beef does not elevate cardiovascular disease risk factors, thus promoting both flavor and health benefits [14,17].

*CAPN1* and *CAST* genes encode the calpain and calpastatin enzymes that regulate protein degradation and meat tenderization post-mortem. *CAPN1* encodes the enzyme μ-calpain, which is a calcium-dependent cysteine protease that breaks down muscle proteins during the meat tenderization process after slaughter. *CAST* encodes the protein calpastatin, which is an endogenous inhibitor of μ-calpain and other calpain enzymes, thereby modulating the extent of protein degradation and meat tenderization. Studies across multiple breeds have validated SNP markers in *CAPN1* (e.g., 316, 530, 4558, 4684) and *CAST* (e.g., 282, 589) as useful for marker-assisted selection to improve beef tenderness [18,19,20].

Polymorphisms in *CAPN1* that beneficially associate with beef tenderness are reported to antagonistically associate with calving day in beef heifers [21] and post-partum interval to estrus in beef cows [22]. However, the results of Cushman et al. [23] indicate that molecular breeding for slice shear force, calculated based on *CAPN1* and calpastatin (*CAST*) genotypes, had minimal or no antagonistic association with reproductive performance in heifers. Table 1 lists some of the commercially available DNA tests for beef quality, although there are more tests in the literature that are being offered to farmers.

The integration of functional mutations in genes such as *MSTN*, *TG5*, *CAPN1*, and *CAST* has led to the development of commercial DNA tests that enhance beef quality traits like tenderness, marbling, and flavor. However, future research should focus on optimizing these genetic advancements alongside animal welfare and environmental factors to ensure sustainable production. Additionally, exploring the interactions between genetic traits and management practices will be crucial for fully realizing the potential of these genomic tools in the beef industry.

## 3. Genome-Wide Association Studies for Beef Quality Traits

Initial genome-wide scans to locate quantitative trait loci (QTL) for beef quality traits were based on linkage analysis within families. For example, Esmailizadeh et al. [1] reported a whole-genome scan to detect QTL for meat quality traits like tenderness (measured as a shear force on two muscles), meat color, pH, and cooking loss, as well as metabolic traits in cattle populations from New Zealand and Australia. The study used backcross calves with Jersey and Limousin backgrounds, with the New Zealand cattle reared on pasture and the Australian cattle finished on grain. A total of 18 significant QTL for meat quality traits and 11 significant QTL for metabolic traits were detected across multiple chromosomes. Genome-wide association studies (GWAS), available since 2005 in human genetics, are based on linkage disequilibrium at the level of a population and involve scanning the entire genome for single nucleotide polymorphisms (SNPs) that are statistically associated with a particular phenotype of interest. GWAS have been successful in the identification of numerous genetic variants associated with complex traits for uncovering novel biological pathways and elucidating the genetic architecture of various traits [25].

Genome association studies provide knowledge about the genetic architecture of beef-related traits that allows linking the target phenotype to genomic information, aiding breeding decisions. GWAS in cattle breeds like Hanwoo (Korean native cattle) have identified 107 significant SNPs on chromosome 14 and candidate genes associated with economically important beef quality traits such as marbling, meat color, texture, and fat color [26]. Nearby genes like SFT2 Domain Containing 3 (*SFT2D3*) and Ectonucleotide Pyrophosphatase/Phosphodiesterase 2 (*ENPP2*) have been highlighted as potential candidate genes affecting beef traits such as marbling and meat color [26].

GWAS results from the study of Forutan et al. [27] implicate some interesting candidate genes (*KIF13A* and *APOB)* for eating quality. Kinesin Family 13A (*KIF13A*) is in a pathway associated with skeletal muscle cells that increases insulin signaling, glucose uptake, and maximal oxygen consumption [28]. Apolipoprotein B (*APOB*) is a building block of a type of lipoprotein called chylomicron. As food is digested, chylomicrons form to carry fat and cholesterol from the intestine into the bloodstream [27].

A recent study [29] performed genome-wide association analyses on Nellore cattle to identify genomic regions and candidate genes influencing carcass traits and meat quality traits (shear force, marbling score, and intramuscular fat content). The top 10 genomic regions explained 8–22% of the additive genetic variance for these traits, harboring a total of 119–155 positional candidate genes. Relevant genes like *CAST*, *PLAG1*, *XKR4*, *PLAGL2*, *AQP3/AQP7*, *MYLK2*, *WWOX*, *CARTPT*, and *PLA2G16* are involved in physiological processes affecting muscle growth, lipid metabolism, adipose tissue development, and signaling pathways like the insulin/IGF-1 pathway.

Mateescu et al. [30] explored the complexity of meat quality by combining GWAS with gene network analysis to identify genes and pathways associated with meat quality traits like tenderness, juiciness, and flavor in Angus cattle. They revealed several modules of co-expressed genes associated with meat quality traits. Key genes identified included *CAST* and *CAPN1* for tenderness, *FASN* and *SCD* for marbling, and *MYOZ1*, *MYOZ3*, and *CASQ1* for color score. The study highlights the utility of network analysis for identifying candidate genes from GWAS results in beef cattle. Several beef cattle studies conducted GWAS to identify genomic regions associated with marbling score, intramuscular fat deposition, and fatty acid composition and revealed several significant SNPs and candidate genes on different chromosomes associated with specific fatty acids and fat content (Table 2).

Genome-wide association and gene enrichment analyses on 672 steers from a multibreed Angus-Brahman beef cattle population have identified membrane anchoring and structural proteins (e.g., ANO2, NTF3, EVC2, ANXA10, PALLD, PKHD1) associated with meat quality traits like tenderness, marbling, cooking loss, and sensory panel ratings for tenderness, juiciness, connective tissue amount, and flavor [32]. A gene network analysis identified *EVC2*, *ANXA10*, and *PKHD1* as potentially harboring multiple QTL for meat quality. The results of Leal-Gutiérrez et al. [32] suggest that polymorphisms in structural proteins can modulate muscle fiber organization and postmortem proteolysis, directly impacting meat quality.

Despite their remarkable success, GWAS have faced several challenges, including the need for larger sample sizes to detect variants with smaller effect sizes and the limited representation of diverse ancestral populations [44]. Additionally, many GWAS are descriptive rather than functionally identifying causal variants. Efforts have been made to increase the diversity of GWAS cohorts and to conduct meta-analyses combining data from multiple studies to enhance statistical power in human genetics [25] and recently in beef cattle [45]. As GWAS continues to evolve, integrating complementary approaches such as functional genomics, epigenomics, and proteomics will be crucial for translating genetic associations into mechanistic insights and understanding the molecular mechanisms underlying beef quality traits.

## 4. Genomic Prediction and Selection for Beef Quality

Genomic selection (GS), which was first introduced by Lande and Thompson [46] and popularized by Meuwissen et al. [47], utilizes genome-wide marker data to predict the so-called genome-enhanced or genomic estimated breeding values (GEBV) of the selection candidates. It involves developing prediction models from a training population with both genotypic and phenotypic data and then applying these models to predict the breeding values of individuals in a separate population based solely on their genotypic information. This approach enables more accurate selection of superior individuals at an early stage, accelerating the rate of genetic gain compared to traditional phenotypic selection. GS relies on capturing the effects of all QTL through linkage disequilibrium between markers and QTL, as well as leveraging genetic relationships between the training and prediction populations [48]. Key factors influencing the accuracy of genomic predictions include the size and genetic diversity of the training population, the heritability of the trait, and the extent of relatedness between the training and prediction sets [48,49]. GS holds the promise to be particularly beneficial in selecting traits such as beef quality traits that are difficult and expensive to measure.

Fernandes Júnior et al. [50] highlighted the long generation interval of beef cattle and the importance of genomic selection in accelerating genetic gains for meat quality traits. Beef tenderness is a significant challenge in the Zebu beef cattle industry. Reported heritability estimates for meat tenderness ranged from 0.11 to 0.45 [51,52]. However, selection for meat quality has only recently (last 10–15 years) been implemented, and due to the long generation interval of beef cattle, substantial genetic improvement is yet to be realized. Additionally, this trait is costly and difficult to measure, and slaughterhouses do not offer differential payment for tender beef. Furthermore, breeding programs have focused more on improving meat quantity over quality attributes. Considering various methods (Bayesian ridge regression, Bayesian LASSO, Bayes A, Bayes B, and Bayes Cπ) and a training population of 426 Nellore animals, Magnabosco et al. [53] reported prediction accuracies for beef tenderness ranging from 0.52 to 0.59. Moderate accuracies for beef tenderness (0.57 to 0.60) have also been reported considering GBLUP, LASSO, and Bayes Cπ in a Nellore training population (n = 4500 animals) [50]. Accuracies between 0.23 and 0.73 were also described by the authors for lipid content, marbling, and meat color (Table 3).

The fatty acid profile is an important indicator of beef quality, and studies have revealed the possibility of genetic improvement of fatty acid composition by selection of both major candidate genes and genomic selection strategies in beef cattle [54,57].

Forutan et al. [27] discussed the use of genomic selection to improve meat quality in beef cattle. They highlighted the shift from producer-driven to consumer-driven beef production and the importance of consumer satisfaction with beef quality. Forutan et al. [27] determined the most accurate method for predicting phenotypes of beef eating quality traits from genotypes and other factors such as carcass weight and days aged. They found that the accuracy of phenotype prediction for beef eating quality traits was sufficiently high that such predictions could be useful in predicting eating quality from samples taken from an animal/carcass as it enters the processing plant to sort for markets with different quality. Forutan et al. [27] emphasized that future predictions should be expanded to incorporate all the parameters in the Meat Standards Australia (MSA) models [58] as well as genotype information.

It has been challenging to implement genomic selection in multi-breed tropical beef cattle populations. If commercial (often crossbred) animals could be used in the reference population for these genomic evaluations, this could allow for very large reference populations. In tropical beef systems, such animals often have no pedigree information. Hayes et al. [59] addressed the challenges of implementing genomic selection in multi-breed tropical beef cattle populations, especially when no pedigree information is available. They evaluated potential models using marker heterozygosity and breed composition derived from genetic markers. The study demonstrated that moderately accurate genomic estimated breeding values (GEBV) can be calculated using these models, with BayesR resulting in the highest accuracy.

The limitations, complexity, and loss of information associated with the multiple-step genomic selection approach [60] have led to the development of single-step approaches [61,62]. Single-step genomic best linear unbiased prediction (ssGBLUP) is a widely used method that combines the pedigree-based numerator relationship matrix (A) and the genomic relationship matrix (G) to construct a combined relationship matrix (H). This allows information from genotyped and non-genotyped individuals to be used simultaneously in one step. The key advantage of single-step methods is that all available information (phenotypic, pedigree, and genomic) is used optimally, leading to greater accuracy and persistence of genomic predictions across generations. It avoids the need for separate evaluations for genotyped and non-genotyped individuals and accounts for potential pre-selection biases. Adekale et al. [63] used the ssGBLUP approach and combined pedigree, genomic, and phenotypic data into one evaluation, and genomic evaluations increased the accuracy of estimated breeding values (EBVs) compared to pedigree-based evaluations alone. They demonstrated the successful implementation of single-step genomic evaluations for improving the accuracy of EBVs in German beef cattle breeding programs across multiple breeds [63].

In summary, challenges in obtaining high-quality and adequately detailed phenotype data, along with frequently incomplete pedigree information, hamper traditional genetic evaluations for beef quality traits. The challenges in collecting beef quality data for genetic evaluations can be attributed to several factors, such as the complexity and variability of the traits being measured, the need for specialized equipment or expertise, and the time and resources required to gather data from a large number of individuals. Additionally, the lack of standardized protocols and the potential for human error in data collection can contribute to the challenges in obtaining high-quality phenotypic data for beef quality traits. Therefore, GS has the potential to substantially increase genetic gain through increased selection accuracy at an early age [64,65]. However, the heterogeneity of breeds, less developed breeding programs and infrastructures, the predominance of natural services, and the population substructures with frequent crossbreeding in commercial herds have restricted the widespread implementation of GS in beef cattle. Multi-breed genomic evaluation and single-step GS are the most recent developments in implementing GS in beef cattle breeding. Challenges include access to large phenotypic datasets across breeds/environments and low-cost genotyping for widespread adoption [66]. Extension of genomic predictions to beef quality traits influencing consumer satisfaction will further require a focus on the collection of reliable phenotypic information across the broad range of traits. Collecting such information will likely rely on public funding efforts. The novel high-throughput phenotyping technologies that facilitate the collection of phenotypes in large cohorts will also be invaluable [66].

## 5. Transcriptomics of Beef Quality

Transcriptomics, one of the most developed fields in the post-genomic era, is the genome-wide study of the complete set of transcribed sequences, including messenger RNA (mRNA), ribosomal RNA (rRNA), transfer RNA (tRNA), and regulatory noncoding RNA in a tissue or a specific cell type at a given time or under a specific physiological condition. Transcriptomics focuses on RNA expression levels to reveal the molecular mechanisms involved in specific biological processes. High-throughput sequencing technologies like bulk RNA-Seq and single-cell RNA-Seq (scRNA-Seq) have transformed transcriptomics research, including studies related to beef quality. Bulk RNA-Seq characterizes average gene expression profiles across samples, enabling the identification of differentially expressed genes and splicing variants associated with meat traits. scRNA-Seq captures cell-type-specific transcriptomes in muscle tissues, revealing cellular heterogeneity and facilitating the discovery of novel cell populations linked to meat quality traits. Together, these complementary high-throughput approaches provide comprehensive insights into transcriptome landscapes and accelerate the development of transcriptome resources for improving beef quality. In addition, the available transcriptomics datasets in cattle, such as the transcriptome atlas [67], can serve as a primary source for biological interpretation and functional validation of transcriptomics studies addressing beef quality complexities.

Intramuscular fat (IMF) deposition has been a central focus of numerous transcriptomics investigations aimed at elucidating the molecular determinants of beef quality [68,69]. A significant proportion of transcriptome research in the realm of beef quality has concentrated on unraveling the genetic and regulatory mechanisms underlying variations in intramuscular fat content, given its pivotal role in influencing meat tenderness, juiciness, and flavor. The study by Yu et al. [69] employed an integrated transcriptomics and metabolomics approach to elucidate the regulatory mechanisms underlying intramuscular fat deposition in three cattle breeds—Qinchuan, Nanyang, and Japanese Black. The Japanese Black breed had significantly higher IMF content compared to the Chinese indigenous breeds. Transcriptomic analysis revealed genes like *ITGB1* were enriched in pathways related to fatty acid metabolism, suggesting their roles in regulating IMF content [69].

Several key regulatory genes have been identified that influence adipocyte differentiation and intramuscular fat deposition, which are important for beef quality. For example, transcription factors like C/EBPα and PPARγ play crucial roles in promoting adipocyte development and fatty acid biosynthesis in beef cattle [68]. Krüppel-like factors (KLFs) are a family of transcription factors that regulate adipogenesis in cattle. KLFs can act as positive or negative regulators of adipocyte differentiation through crosstalk with C/EBP and PPARγ [70]. Adipogenic genes like *DGAT1*, *FABP3*, *FABP4*, and *FASN* are upregulated during early adipocyte differentiation in cattle [71]. In summary, transcription factors like C/EBP, PPARγ, and KLFs, fatty acid metabolism genes, and growth-related genes play key regulatory roles in controlling adipocyte differentiation and intramuscular fat deposition, which are crucial determinants of beef quality. Identifying genetic markers in these pathways can help improve meat quality through breeding programs.

A recent study [72] suggests that long non-coding RNAs (lncRNA) may have critical functional roles in intramuscular fat accumulation. Zhang et al. [72] reported that a lncRNA named long non-coding RNA BNIP3 (lncBNIP3) inhibited the proliferation of bovine intramuscular preadipocytes through the cell cycle pathway, revealing potential new strategies for improving beef quality.

Transcriptomics has been widely exploited to study the effects of diverse feeding systems, production practices, and rearing conditions on beef quality. Researchers have investigated the transcriptomic profiles associated with different dietary regimes, feed restriction and compensatory growth, production systems, and environmental stressors (heat, transportation). These studies aim to elucidate the molecular mechanisms underlying variations in beef quality traits influenced by various production factors. For example, the study by Zhao et al. [73] investigated the effects of acute stress on beef tenderness and the underlying molecular mechanisms in Angus cattle using a functional genomics approach. They found that acute stress significantly increased beef tenderness, as measured by the Warner–Bratzler shear force (WBSF). Microarray analysis identified 147 differentially expressed genes (DEGs) between the stressed and control groups, with the majority of DEGs being downregulated in the stressed group. Functional annotation revealed that these DEGs were enriched in pathways related to muscle structure and integrity, including cytoskeletal organization, muscle contraction, and calcium signaling. Key DEGs included *CAPN1*, *CAPN2*, *CAST*, and *CALM*, which are involved in the calpain-calpastatin system regulating protein degradation and tenderization. The study also identified potential transcriptional regulators, such as NFKB1, CREB1, and FOXO3, that may mediate the stress response and influence beef tenderness. Overall, this functional genomics study provided insights into the molecular mechanisms by which acute stress improves beef tenderness, highlighting the role of the calpain system and related pathways [73]. Sweeney et al. [74] identified 26 differentially expressed (DE) genes related to lipid metabolism between pasture-fed and concentrate-fed cattle. The expression of *ALAD*, *EIF4EBP1*, and *NPNT* could be used to classify the samples based on the production system with 95–100% accuracy [74]. In addition, Deng et al. [75] analyzed the transcriptomes of cattle under varied restricted feeding conditions to study compensatory growth effects on meat quality. Compensatory growth was observed in the restricted groups, accompanied by alterations in meat quality traits like pH, cooking loss, and fat content compared to the ad libitum group. Transcriptome analysis identified DEGs unique to each feeding group as well as shared DEGs involved in pathways related to muscle growth, lipid metabolism, and nutrient utilization. Gene set enrichment analysis further highlighted pathways associated with compensatory growth, such as protein synthesis, cell cycle regulation, and energy metabolism.

The study by Zhang et al. [76] employed comparative transcriptomics to characterize region-specific gene expression patterns across five different beef cuts (tenderloin, longissimus lumborum, rump, neck, chuck) from cattle. They identified a total of 80 region-specific genes (RSGs) and 25 transcription factors regulating these RSGs. Through co-expression network analysis, seven region-specific modules were detected, including three positively and four negatively correlated modules. Their analysis revealed 91 candidate genes associated with meat quality traits, enriched in pathways related to muscle fiber structure, fatty acid metabolism, amino acid metabolism, ion channel binding, protein processing, and energy production. Key genes identified included *TNNI1*, *TNNT1* (muscle structure), *SCD*, *LPL* (fatty acid metabolism), *ALDH2*, *IVD*, *ACADS* (amino acid metabolism), *PHPT1*, *SNTA1*, *SUMO1*, *CNBP* (ion binding), *CDC37*, *GAPDH*, *NRBP1* (protein processing), and *ATP8*, *COX8B*, and *NDUFB6* (energy metabolism) [76]. The differential expression of these RSGs and candidate genes across beef cuts suggests they play a key role in determining region-specific differences in nutrient profiles like fatty acid composition and amino acid content, as well as meat quality traits like tenderness and flavor.

Transcriptomics can provide insights into the molecular mechanisms regulating beef quality traits such as water-holding capacity (WHC). In this regard, Du et al. [77] investigated the molecular mechanisms underlying WHC in Chinese Simmental beef cattle through transcriptome profiling. The longissimus dorsi muscles from 49 cattle were evaluated for meat quality traits, including WHC, water loss, intramuscular fat content, shear force, and pH. Eight individuals with extreme WHC values were selected for RNA-sequencing analysis. A total of 865 DEGs were identified between the high and low WHC groups. These DEGs were involved in pathways related to muscle structure, energy metabolism, and protein folding. The study confirmed seven previously known genes (*HSPA12A*, *HSPA13*, *PPARγ*, *MYL2*, *MYPN*, *TPI*, and *ATP2A1*) and identified six novel candidate genes (*ATP2B4*, *ACTN1*, *ITGAV*, *TGFBR1*, *THBS1*, and *TEK*) potentially affecting WHC [77].

In summary, the recent high-throughput transcriptomic studies have identified differentially expressed genes and pathways involved in lipid metabolism, muscle fiber properties, energy production, and other processes that influence beef quality traits like tenderness, fatty acid composition, and nutrient content across different production systems, feeding regimes, and muscle cuts. This knowledge of the region-specific, breed-specific, and production system-specific gene expression patterns that regulate various aspects of beef quality can guide targeted breeding programs and optimized management practices to improve beef quality.

## 6. Proteomics of Beef Quality

Although transcriptomics tools such as RNA-seq offer a massively parallel approach to genome-wide mRNA expression analysis, there is often no direct relationship between the in vivo concentration of an mRNA and its encoded protein. The association of protein expression levels with biological changes is one of the most fundamental approaches to understanding the functions of individual proteins in complex cellular processes. Proteomics, a large-scale study of proteins, is a biomarker approach for the identification and quantification of all proteins, the proteome, of a given biological system (cell, tissue, organ, biological fluid, or organism) at a specific point in time. Mass spectrometry [78], coupled with advanced separation techniques like two-dimensional gel electrophoresis and liquid chromatography, is the technique most often used for proteomics. In the context of beef quality, proteomics provides insights into the molecular mechanisms influencing meat tenderness, flavor, and other quality attributes. By analyzing the proteome of beef muscles, researchers can identify biomarkers associated with desirable traits, elucidate pathways regulating meat characteristics, and develop strategies to improve beef quality through breeding or processing methods.

Over the last two decades, proteomics has been employed to decipher the underlying factors contributing to variation in beef tenderness. Table 4 summarizes some of the published proteomic studies on beef quality. Functional proteomic analysis was used to associate electrophoretic bands from the myofibrillar muscle fraction with meat tenderness to understand the mechanisms controlling tenderness [79]. Six significant electrophoretic bands were characterized and sequenced, revealing proteins involved in structural, metabolic, chaperone, and developmental functions [79].

An integromics study was performed to review the status of protein biomarker discovery targeting beef tenderness, gathering and proposing a comprehensive list of 124 putative protein biomarkers derived from 28 independent proteomics-based experiments [90]. In the study of Gagaoua et al. [90], 33 robust candidates were identified as worthy of evaluation using targeted or untargeted data-independent acquisition proteomic methods. The study provides an overview of the interconnection of the main biological pathways impacting tenderness determination, including structural proteins, enzymes, heat shock proteins, and proteins involved in energy metabolism, response to oxidative stress, and apoptosis [90]. Gagaoua et al. [90] identified MYOZ3 (Myozenin 3), BIN1 (Bridging Integrator-1), and OGN (Mimecan) as the primary proteins, which accounted for 79% of the variability in shear force values.

Functional proteomic and interactome analysis was used to identify protein biomarkers and biological pathways associated with beef tenderness in Angus cattle [85]. The study compared the proteome of *longissimus thoracis* muscle samples from Angus cattle with divergent tenderness phenotypes. Several proteins involved in structural integrity, energy metabolism, stress response, and proteolysis were found to be differentially abundant between tender and tough meat samples. Interactome analysis revealed complex interactions among these proteins, providing insights into the molecular mechanisms underlying beef tenderness variation. The results of Zhao et al. [85] suggest that a combination of protein biomarkers could be used to predict and improve beef tenderness in Angus cattle. In addition, proteomic techniques have been applied to investigate different degrees of meat tenderness in the Nellore breed, a *Bos indicus* breed of cattle [82,83]. The results demonstrate that meat tenderness in Nellore cattle depends on the modulation and expression of a set of proteins. For example, the results of Rosa et al. [82] demonstrated that polymorphisms at UOGCAST and CAPN4751 SNPs (located on *CAST* and *CAPN1*, respectively) are associated with the variability in the expression of proteins that are involved in muscle metabolism and consequently affect meat tenderness. Malheiros et al. [83] also identified the proteins PFN1, LAP3, PRDX1, PRDX2, HSPD1, and ARHGDIA to be associated with beef tenderness.

The study by López-Pedrouso et al. [91] employed a quantitative proteomic approach using SWATH-MS (Sequential Window Acquisition of All Theoretical Mass Spectra) to investigate the molecular factors influencing beef tenderness in young Piedmontese bulls. They analyzed the proteome of *Longissimus thoracis* muscle samples from 10 animals, which were categorized as tough or tender based on Warner–Bratzler shear force measurements. The SWATH-MS analysis identified and quantified over 1200 proteins, revealing significant differences in the abundance of 43 proteins between the tough and tender groups. Most of these differentially abundant proteins were associated with energy metabolism pathways. Functional analysis suggested that gluconeogenesis, glycolysis, and the citric acid cycle are key pathways influencing tenderness in Piedmontese beef, with proteins like ACO2, MDH1, MDH2, CS, FBP2, PFKL, LDHA, TPI1, and GAPDH/S playing crucial roles [91].

Zhu et al. [80] used label-free proteomics to identify molecular mechanisms and biomarkers related to beef sensory texture and flavor traits in early post-mortem muscle. The authors revealed 34 putative protein biomarkers that discriminated between tender and tough meat groups, belonging to biological pathways associated with muscle structure, heat shock proteins, energy metabolism, response to oxidative stress, and apoptosis. Many of these proteins were previously identified as biomarkers of beef tenderness in an integromics data mining approach [94]. Heat shock protein beta-6 (HSPB6) has been identified as being negatively correlated with tenderness and flavor and positively with stringiness [80]. It belongs to small heat shock proteins (HSPs) that are widely considered useful biomarkers of beef tenderness, color, water-holding capacity, and other quality traits [84,90,95].

To provide insights into the molecular mechanisms underlying dark-cutting beef and identify potential biomarkers for predicting and managing this meat quality defect, Gagaoua et al. [96] conducted an integromics meta-analysis of proteomics data from eight studies on dark-cutting beef. The authors curated a list of 130 proteins that differed between dark-cutting and normal-pH beef and analyzed them using bioinformatics tools. Key pathways involved in dark-cutting beef development included muscle structure, heat shock proteins, energy metabolism, oxidative stress response, and apoptosis. Also, Kiyimba et al. [93] compared the mitochondrial proteomes of dark-cutting and normal-pH beef using LC-MS/MS proteomics and found that dark-cutting beef has up-regulation of proteins involved in mitochondrial biogenesis, oxidative phosphorylation, intracellular protein transport, and calcium homeostasis. Mitochondria isolated from dark-cutting beef showed greater mitochondrial complex II respiration and uncoupled oxidative phosphorylation, but no differences in membrane integrity or respiration at complexes I and IV. These results indicate that dark-cutting beef has greater mitochondrial biogenesis proteins, increasing mitochondrial content and contributing to the dark color. The study provides insights into the mechanistic basis of dark-cutting beef and identifies potential candidate markers for detecting pre-slaughter events leading to this meat quality defect.

In summary, proteomics has been extensively applied to study the molecular basis of various beef quality traits, including tenderness, marbling, color, water-holding capacity, and dark-cutting beef. These studies have utilized advanced proteomics techniques, such as 2D-PAGE, mass spectrometry, and bioinformatics, to identify differentially expressed proteins and their associated biological pathways. Key proteins and pathways linked to meat quality include those involved in glycolysis, oxidative phosphorylation, the tricarboxylic acid (TCA) cycle, muscle structure, heat shock response, energy metabolism, oxidative stress, and apoptosis. Proteomics has provided valuable insights into post-mortem changes in muscle proteins and their relation to the development of meat quality traits, as well as identified potential biomarkers for predicting and managing beef quality. Future research should focus on integrating proteomic analyses with other omics approaches, such as transcriptomics and metabolomics, to gain a more comprehensive view of the regulatory networks influencing beef quality.

## 7. Metabolomics of Beef Quality

Metabolomics is a valuable analytical approach for studying the small-molecule metabolites present in biological samples, including beef and meat products. It utilizes two major platforms, mass spectrometry (MS) and nuclear magnetic resonance (NMR), to comprehensively profile the metabolite composition. The resulting metabolomic data provide insights into the metabolic state of the beef samples, enabling the discovery of biomarkers associated with desirable beef quality traits like tenderness, flavor, and shelf life. Additionally, metabolomics elucidates the underlying biochemical pathways that produce key metabolites influencing beef quality characteristics. Metabolomic profile data can also be used to explore the genes responsible for specific metabolite-featured phenotypes in genome-wide association studies. Therefore, by associating metabolite profiles with sensory evaluation, production conditions, and postmortem changes, metabolomics offers a powerful tool for monitoring and predicting beef quality, optimizing animal breeding and feeding strategies, and improving meat processing methods. Some of the published applications of metabolomics in assessing beef quality are summarized in Table 5.

Muroya et al. [105] introduced the concept of “MEATabolomics”—the application of metabolomics to study skeletal muscle and meat in domestic animals. Muscle metabolites, as the major phenotypic components, determine the physiological characteristics of muscle and meat quality traits. Since raw and cooked meat is rich in flavor-associated volatile compounds and precursors [105], MEATabolomics studies in combination with sensory evaluation can be used to explore biomarker candidates associated with the eating quality of beef.

Jeong et al. [100] used NMR spectroscopy to investigate the meat metabolite profiles related to differences in beef quality attributes, specifically comparing high-marbled and low-marbled groups. High-marbled meat had higher levels of taste compounds compared to low-marbled meat. Metabolite analysis revealed differences between the two marbling groups based on partial least squares discriminant analysis (PLS-DA). Metabolites identified by PLS-DA, such as N, N-dimethylglycine, creatine, lactate, carnosine, carnitine, sn-glycero-3-phosphocholine, betaine, glycine, glucose, alanine, tryptophan, methionine, taurine, and tyrosine, were directly linked to marbling groups. These potential markers were involved in beef taste-related pathways, including carbohydrate and amino acid metabolism. The findings of Jeong et al. [100] provide an important understanding of the roles of taste-related metabolites in beef quality attributes and suggest that metabolomics analysis of taste compounds and meat quality may be a powerful method for evaluating beef quality.

Zhang et al. [106] described recent applications of metabolomics in evaluating meat freshness, composition, authenticity, and origin, highlighting its potential as a powerful tool for meat quality assessment. They discussed the challenges faced, such as sample complexity, a lack of specialized databases, and the need for harmonized methods. Finally, they outlined future trends, including the development of standardized protocols, meat metabolome databases, and advanced data analysis tools to fully exploit the potential of metabolomics in meat science [106]. Moreover, Ramanathan et al. [107] provided a recent comprehensive overview of the current state of metabolomics research in meat quality and highlighted the immense potential of metabolomics in advancing meat quality research and its practical applications in the meat industry.

Yu et al. [69] compared the metabolomes of two Chinese indigenous cattle breeds (Qinchuan and Nanyang) and Japanese Black cattle. They reported that the Japanese Black breed had significantly higher IMF content compared to the Chinese indigenous breeds. Metabolomic analysis showed higher levels of monounsaturated and polyunsaturated fatty acids, as well as amino acids like creatine, lysine, and glutamine, in the Japanese Black breed, contributing to better flavor formation [69].

Metabolomics, especially focusing on volatile compounds, has changed our understanding of beef aroma and flavor. Using techniques like gas chromatography-mass spectrometry (GC-MS), researchers can quantify and correlate metabolites with flavor preferences. This analysis identifies key flavor compounds and their precursors, revealing mechanisms like the Maillard reaction, thermal lipid degradation, and oxidation. In beef, metabolomics shows that flavor results from interactions between aromatics and taste components, with meaty and roasted notes from Maillard reactions [108]. This knowledge helps food scientists predict and manipulate flavor profiles, enhancing product development and quality control.

Castejón et al. [103] investigated the potential of using metabolomics analysis of meat exudate to evaluate beef conservation and aging. These researchers analyzed the exudate from beef samples stored at different temperatures and aging times using NMR spectroscopy. They found that the metabolite profile of the exudate changed significantly with storage temperature and aging time, allowing them to discriminate between fresh and aged meat samples. Specific metabolites like creatine, carnosine, and anserine were identified as potential biomarkers for monitoring meat aging and conservation, demonstrating that metabolomics of meat exudate could be a rapid and non-destructive approach to assessing beef quality during storage and aging processes [103].

Tian et al. [99] performed a comparative metabolomics analysis on subcutaneous fat samples from crossbred cattle with white and yellow fat colors. Through liquid chromatography-mass spectrometry, 235 significant metabolites across five categories were identified, with principal component analysis showing distinct clustering of white and yellow fat samples. White fat exhibited greater metabolite variation, with 163 metabolites having a higher relative abundance and 72 having a lower relative abundance compared to yellow fat. Notably, 3-hydroxyoctanoic acid, anethofuran, 9,10-DiHODE, furanoeremophilane, pregeijerene, N-glycolylneuraminic acid, and glycocholic acid were identified as potential biomarkers for differentiating fat color. The findings provide insights into the metabolic mechanisms underlying fat color variation and suggest potential biomarkers for selective breeding programs aimed at achieving desired beef fat color traits.

Next-generation phenotyping (NGP) using metabolomics is becoming a fundamental approach to refine trait description and improve the prediction of breeding values aligned with beef industry objectives. For example, non-invasive urinary biomarkers have been identified for beef production efficiency and carcass quality traits [101]. These biomarkers are indicative of various aspects of beef quality, such as taste and appearance, that can be used to predict and improve beef quality through targeted breeding and nutrition.

In summary, metabolomics has emerged as a powerful tool for profiling meat quality attributes such as flavor, color, and texture. Recent studies have successfully applied metabolomics to identify biomarkers related to meat quality and taste, using techniques like nuclear magnetic resonance spectroscopy and mass spectrometry. However, challenges remain in correlating metabolites to specific meat quality traits and elucidating the underlying mechanisms [106,107]. Future research should focus on developing generic validation schemes for metabolomics-based meat quality control as well as integrating metabolomics with other omics technologies to provide a more holistic understanding of beef quality.

## 8. Challenges and Future Directions

While a wide range of omics technologies have been applied to study beef quality traits, several challenges remain in fully harnessing their potential:-Integrating Multi-Omics Data: Combining genomics, transcriptomics, proteomics, and metabolomics data to elucidate the complex biological networks underlying meat quality is a challenging task that requires robust bioinformatic pipelines and systems biology approaches.-Implementing Integromics: Integromics, which uses advanced computational and statistical methods to integrate diverse data types, offers a promising platform for advancing beef quality research. However, the implementation of an integromics approach is still in its early stages and requires further development and validation.-Identifying Causal Functional Mutations: The identification and validation of causal functional mutations through gene editing techniques is crucial for precise genomic selection and breeding programs. While gene editing technologies like CRISPR/Cas9 have been developed, their application in beef quality research is still limited.-Overcoming Challenges through Interdisciplinary Research: Addressing the challenges in applying omics technologies to beef quality research will require interdisciplinary research efforts and public-private partnerships. The lack of collaboration between different disciplines and stakeholders has hindered progress in this field.-Translating Multi-Omics Findings into Practical Applications: While multi-omics findings have the potential to improve breeding strategies and genomic predictions for beef quality, the translation of these findings into practical applications is still limited. More research is needed to bridge the gap between research and industry.

Despite these challenges, there have been notable successes, such as the identification of specific genetic variants that have been incorporated into breeding strategies, leading to measurable improvements in meat quality. Future research should focus on refining these techniques, improving data integration methods, and addressing the economic feasibility of implementing functional genomics in commercial cattle breeding. By doing so, we can better harness the potential of these advanced technologies to meet the growing demand for high-quality beef.

## Figures and Tables

**Table 1 genes-15-01104-t001:** A list of some of the commercialized DNA tests for beef quality.

Gene Symbol	Beef Attribute	Discovered by	Commercialized by
*TG*	Marbling	CSIRO/MLA	Genetic Solutions Pty Ltd. (Albion, QLD, Australia)
*CAST*	Meat tenderness	CSIRO/MLA/Beef CRC	Genetic Solutions Pty Ltd.
*CAPN1*	Meat tenderness	USDA/AgResearch NZ	Open
*GH1*	Marbling	NIAS, Japan	Prescribe Genomics Co. (Ibaraki, Japan)
*LEP*	Marbling/fat traits	Univ. of Saskatchewan	Merial, Inc. (Duluth, GA, USA)
Multiple tests	Marbling	-	Genetic Solutions Pty Ltd.
*CAPN3*	Meat tenderness	CSIRO/MLA/Beef CRC	Genetic Solutions Pty Ltd.
*SCD*	Fatty acid composition	Kobe University	Prescribe Genomics CO

The contents of the table were adopted from Hocquette et al. [24] and adjusted for beef quality traits.

**Table 2 genes-15-01104-t002:** Some of the published significant GWAS results for beef quality traits.

Beef Attribute	Population	Sample Size	Genotyping Platform	Significant Genomic Regions/Genes	Reference
Tenderness	Angus cattle	1833	Illumina BovineSNP50 BeadChip	*CAST* and *CAPN1* for tenderness	[30]
Marbling score	Simmental bulls	785	Illumina BovineHD BeadChip	*TUBB1* and *RPL27A* for marbling score	[31]
Warner–Bratzler Shear Force (WBSF), marbling, cooking loss, tenderness, juiciness, connective tissue and flavor	Multibreed Angus–Brahman steers	672	GGP Bovine F-250 chip containing 221,077 SNPs	*LRP5*, *COL3A1*, *GRIP1*, *RECQL5*, *ANO2*, *NTF3*, *CD36*, *GPR98*, *MMRN2* and *GOSR2*.	[32]
Marbling score, meat texture, meat color, and fat color	Hanwoo steers	2110	Illumina Bovine SNP50 BeadChip imputed to a higher density of 15,536,497 SNPs	*SFT2D3* (marbling) located on BTA2, *ENPP2* (meat color) on BTA14, *CPAMD8* on BTA7 and *RHCG* on BTA21 for fat color	[26]
Tenderness, marbling, and flavor, marbling, Warner–Bratzler shear force (WBSF), tenderness, and connective tissue	Angus-sired population of steers, bulls, and cows progeny	2268	Bovine SNP50 Infinium II BeadChip imputed to 44.3 million SNPs	Tenderness: *CAST* and *CAPN1*; WBSF: *CAPN1*, *AGAP1*, *ANXA10*, *CCDC80*, Connective Tissue: *UTRN*, *TMX1*, *TMEM170B*; Marbling: *EGR2*, *RNF130*, *C1QTNF8*, *SOX8*, *SSTR5*, *TEKT4*, *SLC20A2*	[33]
Meat color, purge loss, cooking loss, meat pH, Warner–Bratzler shear force.	Piedmontese young bulls	1166	GeneSeek Genomic Profiler Bovine LD’ (GGP Bovine LD) array containing 30,111 SNPs	SNPs on BTA4 (at ~112.51 Mb), BTA23 (at ~3.91 and ~7.25 Mb), BTA24 (at ~19.87 Mb), and BTA25 (at ~11.96 Mb) for meat color. Water holding capacity: one SNP located on BTA9 (at ~48.33 Mb) for purge loss, and two SNPs located on BTA6 (at ~29.23 Mb) and on BTA10 (at ~14.57 PMb) for cooking loss, one SNP on BTA8 (at ~28.46 Mb) for meat pH.	[34]
Color, aroma, tenderness, juiciness, and palatability	Hanwoo steers	250	Affymetrix Bovine Axiom Array 640K SNP chip	Three pleiotropic SNPs (AX-26703353 and AX-26742891 on BTA6, and AX-18624743 on BTA10) influenced multiple traits like tenderness, juiciness, and palatability	[35]
Oleic acid (C18:1) content in the intramuscular fat	Japanese Black cattle	160	BovineSNP50 BeadChip	A total of 32 SNPs, including the FASN gene, had significant effects on C18:1 levels, with 30 SNPs located between 49 and 55 Mbp on chromosome 19	[36]
Fatty acid composition	Chinese Simmental beef cattle	723	Illumina BovineHD BeadChip	SNPs near the *FASN* gene on BTA19 for C14:0 and C14:1, and the *ELOVL5* gene on BTA23 for C14:0.	[37]
Marbling score and tenderness	Crossbred beef cattle	747	BovineSNP50 BeadChip	One SNP (BTA-60019) on BTA25 accounted for 2.67% of the variation in tenderness.	[38]
Fatty acid composition	Japanese Black cattle	461	BovineSNP50 BeadChip	*FASN* gene on BTA19, one SNP for C18:1 on BTA23, two SNPs for C16:0 on BTA25, and two SNPs for C14:1 near the *SCD* gene on BTA26.	[39]
Fatty acid composition	Angus beef cattle	1713	BovineSNP50 BeadChip	*FASN*, *SCD* and *THRSP* genes	[40]
Intramuscular fat deposition and composition	Nellore steers	585	Illumina BovineHD BeadChip	SNPs near the *FASN* gene on BTA19 for C16:0 and C18:1 fatty acids, and SNPs on BTA7 for intramuscular fat percentage	[41]
Fatty acid composition	American Black Angus calves	2177	574,662 SNPs imputed from BovineSNP50 BeadChip and BovineHD BeadChip	Candidate genes *FABP2*, *FASN*, *FADS2*, *FADS3* and *SCD*	[42]
Fatty acid composition	Nellore cattle	1057	Illumina BovineHD BeadChip	SNPs near the *FASN* gene on BTA19 for C16:0 and C18:1 and the *SCD* gene on BTA26 for C14:1 and C16:1., *THRSP*, *ELOVL6* and *FADS2*	[43]
Eating quality traits: scores for tenderness, juiciness, flavor overall liking	Steers, heifers, and bulls from Brahman, Angus, Hereford, Shorthorn, Holstein, Jersey, Belmont Red, Santa Gertrudis composite, crossbred unknown breed.	1701	709,068 Imputed SNPs from the Illumina HD array	Tenderness: *CAPN1*, *CAST* genes; juiciness and flavor: *MOXD1 APOB*, *KIF13A*	[27]
Shear force, marbling score, and intramuscular fat	Nellore cattle	6910 young bulls with phenotypic information and 23,859 genotyped animals	435,447 Imputed SNPs from multiple Bead chip assay densities	Several candidate genes located on chromosomes BTA1, 2, 5, 7, 9, 10, 19, and 25 for Shear force, on BTA4, 7, 10, 11, 12, 13, 15, and 20 for marbling score, and BTA8, 9, 10, 12, 13, and 28 for intramuscular fat	[29]

BTA: Bovine chromosome.

**Table 3 genes-15-01104-t003:** Genomic prediction accuracies for beef quality traits ^1^.

Trait	Accuracy N		Reference
Meat tenderness	0.52 to 0.59	427	[53]
Meat tenderness	0.57 to 0.60	5062	[54]
Lipids	0.23	3812	[54]
Marbling	0.32	5039	[54]
Carcass intramuscular fat %	0.20	1031	[55]
Marbling score	0.08 to 0.56	4228	[56]
a * color	0.40	5052	[54]
b * color	0.49 to 0.53	5046	[54]
L * color	0.68 to 0.73	5071	[54]
Sum of SFA	0.04 to 0.24	868	[57]
Sum of MUFA	0.05 to 0.13	868	[57]
Sum of PUFA	0.15 to 0.56	868	[57]

^1^ The table was partially adopted from Fernandes Júnior et al. [50]. a *, b *, and L * color refer to the redness, yellowness, and lightness of the meat, respectively. Sum of SFA: Sum of Saturated Fatty Acids, Sum of MUFA: Sum of Monounsaturated Fatty Acids, Sum of PUFA: Sum of Polyunsaturated Fatty Acids.

**Table 4 genes-15-01104-t004:** Summary of some of the published proteomic studies on beef quality.

Beef Attribute	Animal and Age at Slaughter	Sample Size	Protein Extracts	Proteomics Platform	N. of Identified Proteins	Reference
Sensory attributes (tenderness, chewiness, stringiness, and flavor)	Limousin-sired bulls, 16 months	34	Total LD muscle proteins	LC-MS/MS	84	[80]
pH, instrumental color, cooking loss, and WBSF	Immunocastrated F1 Montana-Nellore, heifers + steers, 15 months	16	Myofibrillar and sarcoplasmic proteins	2D-PAGE, MS (ESI-MS/MS)	23	[81]
Tenderness (WBSF)	Nellore cattle, steers, and bulls, 27.7 months	155	Whole LD muscle proteins	2DE and mass spectrometry, MALDI-TOF/TOF MS/MS	40	[82]
Tenderness (WBSF)	Nellore bulls, 27 months		Cytoplasmatic proteins	2D-PAGE, MS (ESI-MS/MS)	29	[83]
pH, WBSF, and WHC	Angus × Simmental beef cattle (USDA Select; A maturity)	8	Whole muscle protein	Western blots, SDS-PAGE	14	[84]
Beef tenderness	Angus Steers, 18 months	6	Myofibrillar proteins	1DE + nano-LC-MS/MS	19	[79]
Beef tenderness	Angus Steers, 12 months	19	High salt and low salt soluble proteins	1DE + LC-MS/MS	8	[85]
Beef tenderness	Angus Steers	15	Myofibrillar and sarcoplasmic proteins	2D-DiGE + Linear Ion Trap MS	28	[86]
Beef tenderness and intramuscular fat	Nellore Bulls + Steers	12	Myofibrillar and sarcoplasmic proteins	2DE + MALDI-TOF/TOF	9	[87]
Beef tenderness	Charolais × Aubrac Heifers, 33 ± 3 months	10	Myofibrillar and sarcoplasmic proteins	Label-free + Nano-LC-MS/MS	40	[88]
Beef tenderness	Charolais Bulls, 17 months	8	Myofibrillar and sarcoplasmic proteins	2DE + MALDI-TOF/TOF	23	[89]
Beef tenderness and marbling	PDO Maine Anjou Cows, 67.4 months	188	Myofibrillar and sarcoplasmic proteins	RPPA	10	[90]
Tenderness (shear force)	Piedmontese bulls, 7 months	10	Cytoplasmatic proteins	SWATH-MS	43	[91]
Dark-cutting	6 dark-cutters and 6 normal-pH beef, other information not visible	12	Label-free quantitative proteomics using LC-MS/MS	Total protein extract	57	[92]
Dark-cutting	Beef cattle, other information not visible	22	LD muscle mitochondrial proteins	LC-MS/MS	12	[93]

WHC: water-holding capacity, WBSF: Warner–Bratzler shear force, 2D-PAGE: Two-dimensional electrophoresis, MS: Mass spectrometry. ESI–MS/MS: Electrospray ionization-tandem mass spectrometry, SWATH-MS: Sequential Windowed Acquisition of All Theoretical Fragment Ion Mass Spectra, LC-MS/MS: Label-free shotgun proteomics combined with liquid chromatography-tandem mass spectrometry.

**Table 5 genes-15-01104-t005:** Summary of some of the applications of metabolomics in beef quality analysis.

Beef Attribute	Analytical Techniques	Multivariate Analysis Techniques	Metabolites	Reference
Sensory evaluation of beef taste	GC/MS	PCA	Cold storage led to increased free fatty acids and Glutamic acid and decreased creatinine and inosinic acid	[97]
Meat color, pH, water holding capacity, shear force, and texture	NMR	PCA, OPLS-DA	Beef quality differences related to acetylcholine, valine, adenine, leucine, phosphocreatine, β-hydroxypyruvate, ethanol, adenosine diphosphate, creatine, acetylcholine, and lactate	[98]
Beef fat color	LC-MS	PCA, PLS-DA	3-hydroxyoctanoic acid, anethofuran, 9,10-DiHODE, furanoeremophilane, pregeijerene, N-glycolylneuraminic acid, and glycocholic acid were identified as potential biomarkers for differentiating fat color	[99]
Marbling	NMR	PLS-DA	Carnosine, creatine, glucose, and lactate were associated with higher marbling	[100]
Marbling	Mass spectrometry-based untargeted and targeted metabolomics	ASCA	Unconjugated-BA and Glucocorticoids were associated with marbling	[101]
Aroma of cooked beef	SPME and GC–MS	Linear and logarithmic regression model	Benzeneacetaldehyde and Heterocyclic compounds	[102]
Meat freshness	NMR	PCA, PLS	60 identified metabolites, metabolomics classified meat samples according to their storage time	[103]
Intramuscular fat	NMR	PCA, OPLS-DA	The unsaturation degree of triacylglycerol was estimated by the ^1^H NMR spectra and was correlated with the content ratio of unsaturated fatty acids and the melting point of IMF. Leucine and creatine were found as biomarkers, positively and negatively correlated with aging duration, respectively.	[104]

NMR: Magnetic resonance spectroscopy, PCA: Principal Component Analysis, PLS-DA: Partial least squares discriminant analysis, SPME: Solid-phase microextraction, and GC–MS: Gas chromatography-mass spectrometry, LC-MS: Liquid chromatography-mass spectrometry, OPLS-DA: Orthogonal signal correction–projection to latent structures–discriminant analysis, ASCA: ANOVA-simultaneous component analysis.

## Data Availability

The original contributions presented in this review are included in the article within the text; further inquiries can be directed to the corresponding author.

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
