# Peer review of "Leveraging Functional Genomics for Understanding Beef Quality Complexities and Breeding Beef Cattle for Improved Meat Quality"

_genes, 2024, doi:10.3390/genes15081104_

Round 1
Reviewer 1 Report
Comments and Suggestions for Authors
The authors reveal enthusiasm rather than a critical review. I would be delighted if there were some specific and useful conclusions.
CRISPR/Cas9 has not enhanced. This is totally wrong.
Why are there "difficulties collecting phenotypic data"? If so, there is little to say.
"GWAS ..have revolutionised ..beef quality traits" is, at best, wishful thinking.
The reality is that the authors have concentrated on SNPs and have not considered the more informative literature
eg SBSmith, T Gotoh
Hints:
1 SCD is widely used. Contrast with "the four key genes"
2 haplotypes of the multimegabase C19 cluster (including SREBP to FASN) are not revealed by the approaches they describe.
Comments on the Quality of English Language
too much hyperbole
Author Response
We would like to thank this reviewer for the useful comments that led to the improvement of our manuscript.
Comments 1: The authors reveal enthusiasm rather than a critical review. I would be delighted if there were some specific and useful conclusions.
Response 1: Thank you for pointing this out. We appreciate your feedback and acknowledge the importance of providing a balanced and critical perspective in our review article. While the aim was to highlight recent advancements in the field, we understand that this may have led to a more enthusiastic tone. In response to your comments, we have revised the manuscript to include specific conclusions in each section that emphasize both the strengths and limitations of the studies discussed. Specifically, we add the subsection entitled “Challenges and Future Directions”
Comments 2: CRISPR/Cas9 has not enhanced. This is totally wrong.
Response 2: Agree. We replace the statement by this ”While gene editing technologies like CRISPR/Cas9 have been developed, their application in beef quality research is still limited.”
Comments 3: Why are there "difficulties collecting phenotypic data"? If so, there is little to say.
Response 3: We revised this. Please see Page: 10 Line 5-13.
Comments 4: "GWAS ..have revolutionised ..beef quality traits" is, at best, wishful thinking.
Response 4: We agree. The referee raises a valid point that the statement "GWAS have revolutionized our understanding of the genetic architecture of beef quality traits" is an overstatement. While these methods have contributed to our knowledge, their impact on revolutionizing our understanding is still a matter of ongoing research and debate within the scientific community. More research is needed to fully validate the findings and translate them into practical applications for improving beef quality. We revised this section. Please see Page 2, Lines: 10-14.
Comments 5: The reality is that the authors have concentrated on SNPs and have not considered the more informative literature
eg SBSmith, T Gotoh
Hints:
1 SCD is widely used. Contrast with "the four key genes"
2 haplotypes of the multimegabase C19 cluster (including SREBP to FASN) are not revealed by the approaches they describe.
Response 5: We appreciate the referee's observation regarding our focus on SNPs. We agree that a broader consideration of the literature would enhance our manuscript. However, we would like to clarify that much of the existing literature primarily discusses nearby genes and SNPs in relation to beef quality traits. In our study, we aimed to identify and report the causal SNPs that are most relevant to our research objectives. We conducted a thorough literature search, and while we found valuable insights, the emphasis on SNPs in the context of beef quality traits remains prevalent. We agree about the “SCD”. We found there was an error and fixed it in Table 2. Thank you for your constructive feedback.
Reviewer 2 Report
Comments and Suggestions for Authors
The lines are not numbered, which makes the manuscript difficult to review.
Abstract
Change to “…state of applications of the omics technologies…”.
Instructions for authors:
Abstract: The abstract should be a total of about 200 words maximum. The abstract should be a single paragraph and should follow the style of structured abstracts, but without headings: 1) Background: Place the question addressed in a broad context and highlight the purpose of the study; 2) Methods: Describe briefly the main methods or treatments applied. Include any relevant preregistration numbers and species and strains of any animals used; 3) Results: Summarize the article's main findings; and 4) Conclusion: Indicate the main conclusions or interpretations. The abstract should be an objective representation of the article: it must not contain results that are not presented and substantiated in the main text and should not exaggerate the main conclusions.
Please check to make sure that your Abstract contains all the required components.
3. Genome-wide association studies for beef quality traits
Is GWAS singular or plural? In one place, you say “GWAS have,” and in another place, you say “GWAS has.”
Change to “…with skeletal muscle cells that increase insulin signaling…”.
Change to “Despite its remarkable success, GWAS has faced…”.
“GWAS has” indicates GWAS is singular. “GWAS are” indicates GWAS is plural. Please be consistent.
Table 2: Leave a space after “of”. Change “meat col” to “meat color”. What is meant by “containing 30
111 SNPs”? Why is 2177 in the Population column?
4. Genomic Prediction and Selection for Beef Quality
Provide references for the statement “Reported heritability estimates for meat tenderness ranged from 0.11 to 0.45”.
Table 3: Move N above the correct column. Define the acronyms in the footnotes to the table.
5. Transcriptomics of Beef Quality
The word “significant” does not need to be capitalized.
Change to “The expression of ALAD, EIF4EBP1 and NPNT could be used to classify the samples based on the production system with 95-100% accuracy (Sweeney et al., 2016)”.
Change to “…acid composition and amino acid content, as well as meat quality traits…”.
6. Proteomics of Beef Quality
“proteomics” does not need to be capitalized.
Table 4: “Loins collected at the same time from commercial abattoir.” Collected at the same time as what? “Sequential Windowed Acquisition of All Theoretical Fragment Ion Mass Spectra” does not need to be in bold type.
Change to “Gagaoua et al. (2020a) identified MYOZ3 (Myozenin 3), BIN1 (Bridging Integrator-1), and OGN (Mimecan) as the primary proteins, which accounted for 79% of the variability in shear force values”.
Change to “…tenderness in the Nellore breed, a Bos indicus breed of cattle…”.
Change to “…SNPs (located on CAST and CAPN1, respectively) are associated with variability in the expression of proteins that are involved in muscle metabolism…”.
Change to “…influencing beef tenderness in young Piedmontese bulls”.
“10” in place of “ten”.
Correct the spelling of “Piedmontese”.
7. Metabolomics of Beef Quality
Change to “explore the genes responsible for…”.
Change to “The resulting metabolomic data provide insights into the metabolic state…” since “data” is plural.
Table 5: Change to “Leucine and…”. Capitalize the first word of the sentence. In the footnote, change to “partial least squares”.
Change to “…partial least squares discriminant analysis…”.
Delete the comma in “Zhang et al., (2021) described recent applications of metabolomics…”.
Comments on the Quality of English LanguageThe manuscript is well-written and well-organized. Only minor editing is needed.
Author Response
We would like to thank this reviewer for the useful comments that led to the improvement of our manuscript.
Comments 1: The lines are not numbered, which makes the manuscript difficult to review.
Response 1: Thank you for pointing this out. We used the journal template which had no line numbered. Please see the revised manuscript in which we put the line numbers.
Comments 2: Abstract: Change to “…state of applications of the omics technologies…”.
Response 2: Yes, we have changed it. Please see the abstract Line 22.
Comments 3: Instructions for authors: Abstract: The abstract should be a total of about 200 words maximum. The abstract should be a single paragraph and should follow the style of structured abstracts, but without headings: 1) Background: Place the question addressed in a broad context and highlight the purpose of the study; 2) Methods: Describe briefly the main methods or treatments applied. Include any relevant preregistration numbers and species and strains of any animals used; 3) Results: Summarize the article's main findings; and 4) Conclusion: Indicate the main conclusions or interpretations. The abstract should be an objective representation of the article: it must not contain results that are not presented and substantiated in the main text and should not exaggerate the main conclusions.
Please check to make sure that your Abstract contains all the required components.
Response 3: We agree that the Abstract should have this structure for a research article but our manuscript is a review with no specific Material and methods or results so we followed the journal instructions for the format of a review article.
Comments 4:
- Genome-wide association studies for beef quality traits
Is GWAS singular or plural? In one place, you say “GWAS have,” and in another place, you say “GWAS has.”
Response 4: Thanks for this point. GWAS is plural. We changed the manuscript in Page: 4, Line:16; Page: 5, Line:4; Page: 4, Line:10; Page: 4, Line:16.
Comments 5: Change to “…with skeletal muscle cells that increase insulin signaling…”.
Response 5: Agree, we change it. Please see Page: 4, Line: 24.
Comments 6: change to “Despite its remarkable success, GWAS has faced…”.
Response 6: Yes, we used GWAS as plural and changed the verbs accordingly. Please see Page: 5, Line: 4.
Comments 7: Table 2: Leave a space after “of”. Change “meat col” to “meat color”. What is meant by “containing 30.
Response 7: Yes, we changed it please see Table 2. It is 30111. We revised it now.
Comments 8: 111 SNPs”? Why is 2177 in the Population column?
Response 8: Thanks. We removed “111” and corrected it. 2177 is correct. We revised it. please see Table 2.
Comments 9:
- Genomic Prediction and Selection for Beef Quality
Provide references for the statement “Reported heritability estimates for meat tenderness ranged from 0.11 to 0.45”.
Response 9: Yes, we added these related references on Page: 8, Lines: 21-22.
Wheeler, T.L.; Cundiff, L.V.; Shackelford, S.D.; Koohmaraie, M. Characterization of biological types of cattle (Cycle VIII): Carcass, yield, and longissimus palatability traits. J. Anim. Sci. 2010, 88, 3070-3083. doi: 10.2527/jas.2009-2497.
Gordo, D.G.M.; Espigolan, R.; Bresolin, T.; Fernandes Júnior, G.A.; Magalhães, A.F.B.; Braz, C.U.; Fernandes, W.B.; Baldi, F.; Albuquerque, L.G. Genetic analysis of carcass and meat quality traits in Nelore cattle. J. Anim. Sci. 2018, 96, 3558-3564. doi: 10.1093/jas/sky228.
Comments 10: Table 3: Move N above the correct column. Define the acronyms in the footnotes to the table.
Response 10: Yes, Thanks. We fixed it. Also, we defined the acronyms in the footnotes to the table. . a*, b* and L* color refer to the redness, yellowness, and lightness of the meat, respectively. Sum of SFA: Sum of Saturated Fatty Acids, Sum of MUFA : Sum of Monounsaturated Fatty Acids, Sum of PUFA: Sum of Polyunsaturated Fatty Acids.
Comments 11:
- Transcriptomics of Beef Quality
The word “significant” does not need to be capitalized.
Change to “The expression of ALAD, EIF4EBP1 and NPNT could be used to classify the samples based on the production system with 95-100% accuracy (Sweeney et al., 2016)”.
Change to “…acid composition and amino acid content, as well as meat quality traits…”.
Response 11: Thank you. The word “significant” was fixed. We Changed to “The expression of ALAD, EIF4EBP1 and NPNT could be used to classify the samples based on the production system with 95-100% accuracy (Sweeney et al., 2016)”. Page 11, Line 39. We changed to “…acid composition and amino acid content, as well as meat quality traits…”. Page 12, Line 10.
Comments 12:
- Proteomics of Beef Quality
“proteomics” does not need to be capitalized.
Table 4: “Loins collected at the same time from commercial abattoir.” Collected at the same time as what? “Sequential Windowed Acquisition of All Theoretical Fragment Ion Mass Spectra” does not need to be in bold type.
Response 12: Yes, we fixed “proteomics”. We removed this “Loins collected at the same time from commercial abattoir.” We added “6 dark-cutters and 6 normal-pH beef, other information not visible”. We fixed this “Sequential Windowed Acquisition of All Theoretical Fragment Ion Mass Spectra”. Please see Table 4.
Comments 13:
Change to “Gagaoua et al. (2020a) identified MYOZ3 (Myozenin 3), BIN1 (Bridging Integrator-1), and OGN (Mimecan) as the primary proteins, which accounted for 79% of the variability in shear force values”.
Response 13: Agree. We revised this please see Page 15, Lines 11-16.
Comments 14: Change to “…tenderness in the Nellore breed, a Bos indicus breed of cattle…”.
Response 14: Yes, we revised this. Please see Page 15, Line 27.
Comments 15: Change to “…SNPs (located on CAST and CAPN1, respectively) are associated with variability in the expression of proteins that are involved in muscle metabolism…”.
Response 15: Yes, we revised this. Please see Page 16, Lines 1-2.
Comments 16:
Change to “…influencing beef tenderness in young Piedmontese bulls”.
“10” in place of “ten”.
Correct the spelling of “Piedmontese”.
Response 16: Yes, we revised this. Please see Page 16, Lines 7-8; Lines 7-8, And Table 4.
Comments 17:
- Metabolomics of Beef Quality
Change to “explore the genes responsible for…”.
Response 17: yes, We changed it.
Comments 18: Change to “The resulting metabolomic data provide insights into the metabolic state…” since “data” is plural.
Response 18: yes, we changed it.
Comments 19:
Table 5: Change to “Leucine and…”. Capitalize the first word of the sentence. In the footnote, change to “partial least squares”.
Change to “…partial least squares discriminant analysis…”.
Response 19: yes, we changed the. Please see Table 5.
Comments 20: Delete the comma in “Zhang et al., (2021) described recent applications of metabolomics…”.
Response 20: yes, We deleted it.
Round 2
Reviewer 1 Report
Comments and Suggestions for Authors
I appreciate the attempts to reduce the hyperbole but the concessions are
insufficient. The paper could be summarised as showing that the various approaches yield countless candidates but no understanding and little prospect for improvements in meat quality. However, patently, selection has been very successful in Japan and Korea and now elsewhere. How could this be?
There are huge databases with all the phenotypic data required.
As previously suggested, the authors should start with some understanding of the history and biology eg Gotoh, Smith and build from there. Many factors affect expression of true snow flake marbling as opposed to some measurable IMF. For example, age in excess of 2 years is critical but often ignored, as in this study.
The authors should consider the actual locations of polymorphic genes including the cluster contained within C19 haplotypes. Black Wagyu have specific haplotypes.
Author Response
We would like to thank this reviewer for the useful comments that led to the improvement of our manuscript.
Comments 1:
The paper could be summarised as showing that the various approaches yield countless candidates but no understanding and little prospect for improvements in meat quality. However, patently, selection has been very successful in Japan and Korea and now elsewhere. How could this be?
There are huge databases with all the phenotypic data required.
As previously suggested, the authors should start with some understanding of the history and biology eg Gotoh, Smith and build from there. Many factors affect expression of true snow flake marbling as opposed to some measurable IMF. For example, age in excess of 2 years is critical but often ignored, as in this study.
The authors should consider the actual locations of polymorphic genes including the cluster contained within C19 haplotypes. Black Wagyu have specific haplotypes.
Response 1: We tried to attend this comment and added materials from Gotoh, Smith. Please see Page 3, Lines: 14-28 and references: Page 22, Line: 32-40. We did not fully understand the reviewer's comments regarding the C19 haplotypes."